# Probabilistic Fusion Approach for Robust Battery Prognostics

**Jokin Alcibar**
Mondragon Unibertsitatea
jalcibar@mondragon.edu

**Ekhi Zugasti**
University of the Basque Country (UPV/EHU)
ekhi.zugasti@ehu.es

**Aitor Aguirre-Ortuzar**
Mondragon Unibertsitatea
aaguirre@mondragon.edu

**Jose I. Aizpurua**
University of the Basque Country (UPV/EHU)
Ikerbasque, Basque Foundation for Science
joxe.aizpurua@ehu.eus

## Abstract

Batteries play a critical role for advancing the decarbonization of the transportation and energy systems. Ensuring their safe and reliable operation is essential for the effectiveness of battery-powered systems. To support this, the development of accurate and robust prognostic models for battery state-of-health is key, enabling autonomous systems to operate effectively in complex and remote settings. The combination of Neural Networks, Bayesian modelling concepts and ensemble learning strategies, form a valuable prognostics framework to combine uncertainty in a robust and accurate manner. Accordingly, this study presents a Bayesian ensemble learning methodology for predicting the capacity degradation of lithium-ion batteries. The approach effectively forecasts capacity fade and quantifies the uncertainty associated with battery design and degradation processes. The proposed methodology employs a stacking ensemble technique, integrating multiple Bayesian Neural Networks (BNNs) as base learners, which have been trained on data diversity. Validation was performed using a battery aging dataset from NASA Ames Prognostics Center of Excellence. Obtained results highlight the enhanced accuracy and reliability of the proposed probabilistic fusion approach compared to (i) pseudo-Bayesian model averaging, (ii) pseudo-Bayesian model averaging with Bayesian bootstrapping, and (iii) a point prediction stacking approach using distinct BNNs.

## 1 Introduction

Batteries are crucial for a sustainable, carbon-free economy. The development of accurate battery Remaining Useful Life (RUL) prediction models is particularly important for reliable energy strategies and cost-effective solutions. The estimation of the state-of-health (SOH) is a key activity for the design of RUL prognostics models, focusing on capturing aging dynamics and health state (Toughzaoui et al., 2022). SOH-based prognostics capture battery ageing and health estimation which are crucial indicators for addressing the degradation that impacts capacity and increases safety risks like overheating (Wang et al., 2022). Thus, precise SOH monitoring and forecasting are imperative for safe and efficient operation of battery-operated systems (Zhao et al., 2023).

Recent data-driven approaches have focused on modeling the capacity degradation of lithium-ion (Li-ion) batteries. Toughzaoui et al. (2022) developed a Convolutional Neural Network (CNN)-Long short-term memory (LSTM) architecture, and Wei and Wu (2023) presented a graph CNN

Workshop on Bayesian Decision-making and Uncertainty, 38th Conference on Neural Information Processing Systems (NeurIPS 2024).

complemented by dual attention mechanisms for the estimation of SOH and RUL of batteries. Due to the variability inherent in the battery manufacturing process, it is essential to quantify this uncertainty to ensure robust and reliable prognostics predictions Abdar et al. (2021); Nemani et al. (2023).

In the broader machine learning context, ensembles of probabilistic models have been utilized to capture complex uncertainties. Fan et al. (2017) introduced a Bayesian posterior predictive framework for weighting ensemble climate models. Cobb et al. (2019) present a new machine learning retrieval method based on an ensemble of BNNs. In this scenario, the overall output from the ensemble is treated as a Gaussian mixture model. However, models are equally weighted with no adaptation to the observed data. Zhang et al. (2022) present a Bayesian Mixture Neural Network (BMNN) for Li-ion battery RUL prediction. However, the absence of a weighted model combination limits the analysis of individual model contributions. Alternatively, Bai and Chandra (2023) described a Bayesian ensemble framework using gradient boosting and Markov Chain Monte Carlo sampling, while Dai et al. (2023) demonstrated a robust Bayesian fusion method within a sequential Monte Carlo algorithm, enhancing uncertainty quantification in predictions.

In this context, inspired by the promising ability of probabilistic ensemble models to capture model uncertainty, the main contribution of this research is the development of a probabilistic model fusion approach for battery SOH predictions. Bayesian convolutional neural networks (BCNNs) are used as base models for SOH prediction, and the developed fusion approach integrates individual BCNN probabilistic predictions. The fusion strategy balances between precision and reliability of individual predictions, adopting an optimal tradeoff between accuracy and uncertainty of predictions through the proposed stacking approach. The proposed approach has been compared with (i) stacking BCNN models using mean information, (ii) Pseudo BMA, and (iii) a Pseudo-BMA variant stabilized with Bayesian bootstrap, called Pseudo-BMA+. Obtained results confirm that the proposed framework infers accurate, well-calibrated, and reliable probabilistic predictions, which improve predictive performance and contribute to estimate uncertainty in a robust and reliable manner in complex data-driven tasks. The proposed approach has been tested and validated with the publicly available NASA's battery dataset Saha and Goebel (2007).

## 2  Methodology

The proposed framework integrates BCNNs with probabilistic ensemble strategies to generate accurate predictions with robust uncertainty quantification, leveraging Bayesian modeling and ensemble strategies. The approach is divided into offline and online stages. From battery datasets, the offline process completes data pre-processing and model training. In the online process, trained models are stacked according to computed weight and stacking criteria. The outcome is a one-step-ahead probabilistic capacity estimate. Figure 1 shows the high-level block diagram of the proposed approach. The approach is generally divided into offline and online phases.

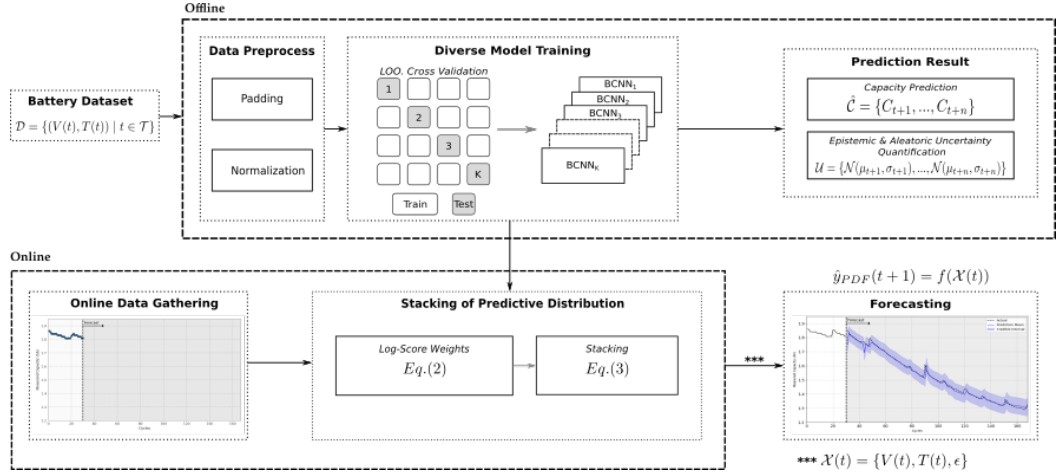

Figure 1: Block diagram of the proposed approach Alcibar et al. (2024).

## 2.1 Offline Phase

During the offline phase, starting from a battery dataset with different run-to-failure trajectories on the same type of batteries, different base models are designed through a training strategy which seeks diversity in the training set to develop complementary predictive models.

### Ensemble Base Models: BCNNs

BCNN combines feature extraction capabilities of classical CNN models with the uncertainty quantification of Bayesian theory. This fusion enables robust predictions by capturing model uncertainty, crucial for reliable decision-making in complex scenarios. The proposed architecture is built using variational layers of `TensorFlow Probability` in Python Dillon et al. (2017). Detailed architecture is provided in the Appendix A.

### Training for Diversity

Model diversity is crucial for effective ensemble models Nam et al. (2021). Accordingly, the training set for each battery model is modified to learn different battery aging properties. Using the leave-one-out (LOO) strategy, $K$ diverse BCNN models are built from $K$ run-to-failure trajectories, changing the training set in each iteration (cf. Figure 1). Each model is trained on all batteries except one, held as a test set. This LOO approach enhances models' ability to generalize across battery types and manufacturing conditions. This stage completes the offline training process, which results in a set of BCNN models, $\mathcal{M} = \{BCNN_1, BCNN_2, \ldots, BCNN_K\}$.

## 2.2 Online Phase: Stacking of Predictive Distribution (SPD)

During the online phase, the proposed stacking of predictive distribution strategy is designed and tested. The proposed approach takes as input individual base models [cf. Eq. (2.1)] and monitored data up to the prediction instant $t$, which is used to forecast the probability density function (PDF) of the capacity at $t+1$, $\hat{y}_{PDF}(t+1)$. The objective of the stacking process is to integrate the predictive distributions of different base models and propagate all the information end-to-end.

For comparison and benchmarking purposes, we implemented alternative ensemble methods: Pseudo-Bayesian Model Averaging (Pseudo-BMA) (cf. Appendix C.1), an enhanced variant of Pseudo-BMA stabilized using Bayesian bootstrap, known as Pseudo-BMA+ (cf. Appendix C.2) and Stacking Point Prediction (SPP) (cf. Appendix C.3).

### Stacking using Logarithmic Score

The optimal way to combine a set of Bayesian posterior predictive distributions is by using the logarithmic score Yao et al. (2018). This method maximizes the average log-likelihood of the observed data, which is a proper scoring rule used to evaluate the accuracy of probabilistic forecasts. It measures the accuracy of a forecast and penalizes overconfidence and underconfidence in the predicted probability. The logarithmic score is defined as follows:

$$\hat{w} = \arg\max_w \frac{1}{N} \sum_{i=1}^{N} log \sum_{k=1}^{K} w_k p(y_i \mid y_{-i}, M_k) + \lambda_{reg} \sum_{k=1}^{K} w_k^2 \tag{1}$$

where $N$ denotes the total number of data points and $K$ denotes the total number of base models. The LOO predictive distribution for each model, *i.e.* $p(y_i \mid y_{-i}, M_k)$, is used to compute the model's prediction for the data point $i$. To avoid overfitting, a regularization term $\lambda_{reg}$ is added.

In the Bayesian framework, stacking extends beyond averaging point predictions by combining multiple Bayesian posterior predictive distributions. This approach develops a *stacking model* that leverages the strengths of various predictive models [cf. Eq. (2.1)], enhancing the overall predictive accuracy. Stacking predictive distributions enables fusion of uncertainties from various models into a unified predictive framework. This approach improves forecast accuracy and offers a comprehensive evaluation of uncertainty associated with forecasts, providing advantages across diverse decision-making scenarios. The fundamental equation governing this process is defined as follows:

$$\hat{p}(\tilde{y}|y) = \sum_{k=1}^{K} \hat{w}_k p(\tilde{y}|y, M_k) \tag{2}$$

where $\hat{p}(\tilde{y}|y)$ represents the aggregate probability estimation based on the ensemble model, $\omega_k$ denotes the weight assigned to the $k$-th component within the ensemble, and $p(\tilde{y}|y, M_k)$ refers to the probabilistic forecast generated by each base model.

## 2.3 Forecasting

Online forecasting computes one-step ahead predictions. To forecast battery capacity at instant $t + 1$, previous data until instant $t$ is used, plus an uncertainty factor expressed as noise. This data includes voltage $V(t)$ and temperature $T(t)$ at instant $t$, as well as a Gaussian noise term $\epsilon$ distributed as $N(0, \sigma = 1)$, which introduces variability in the data's progression over time. The one-step ahead capacity distribution prediction is thus defined as :

$$\hat{y}_{PDF}(t + 1) = f(\mathcal{X}(t)) \tag{3}$$

where $f(.)$ denotes the designed ensemble model, and $\hat{y}_{PDF}(t + 1)$ is the distribution of the capacity estimate at $t + 1$.

## 3 Results

To evaluate the proposed approach, different ensemble strategies are compared to evaluate their strengths and identify the most suitable approach. Table 1 presents a comparative analysis in terms of accuracy and probabilistic metrics.

A notable observation from the results in Table 1 is the variance between the proposed ensemble approach (cf. Figure 1) and the benchmarking ensemble models (cf. Appendix C) in specific scenarios. For batteries #5, #6 and #7, the proposed approach exhibited superior outcomes, particularly in probabilistic metrics such as Negative Log-Likelihood (NLL) (cf. Appendix D.2) and Continuous Rank Probability Score (CRPS) (cf. Appendix D.1). This suggests that the method produces not only accurate point estimates but also well-calibrated probability distributions. This is particularly valuable for battery health prognostics, where understanding prediction uncertainty is crucial for informed decision-making.

Table 1: Comparison of different ensemble strategies for different batteries used as test.

| | Pseudo-BMA | | Pseudo-BMA+ | | SPP | | SPD | |
|---|---|---|---|---|---|---|---|---|
| | $NLL(\downarrow)$ | $CRPS(\downarrow)$ | $NLL(\downarrow)$ | $CRPS(\downarrow)$ | $NLL(\downarrow)$ | $CRPS(\downarrow)$ | $NLL(\downarrow)$ | $CRPS(\downarrow)$ |
| B0005 | -2.141 | 0.014 | -2.158 | 0.013 | -1.667 | 0.025 | **-2.163** | **0.012** |
| B0006 | -1.973 | 0.018 | -1.965 | 0.018 | -1.709 | 0.024 | **-2.008** | **0.016** |
| B0007 | -2.172 | 0.013 | -2.181 | 0.013 | -1.986 | 0.015 | **-2.191** | **0.012** |
| B0018 | -1.493 | 0.031 | -1.553 | 0.029 | **-1.799** | **0.019** | -1.593 | 0.0286 |

Figure 2 displays a visual comparative analysis corresponding to the data presented in Table 1, showcasing the different ensemble methods used for forecasting the capacity degradation of battery #5. The pseudo-BMA method displays a relatively narrow credible interval, though its accuracy is not very precise, especially at the beginning. The pseudo-BMA+ method shows similar uncertainty quantification to pseudo-BMA, with slightly improved accuracy. In contrast, the stacking of point predictions method yields the least favorable results in both accuracy and uncertainty quantification. Notably, the stacking of predictive distributions method provides the most comprehensive results, combining high accuracy with narrow credible intervals.

These observations highlight that although pseudo-BMA and pseudo-BMA+ present good results, the stacking of predictive distributions is generally superior due to its connection with the logarithmic score. This approach avoids potential issues such as identification problems arising from the lack of strict propriety associated with the energy score, and eliminates the need for additional smoothness assumptions required by other proper local scoring rules (Gneiting and Raftery, 2007).

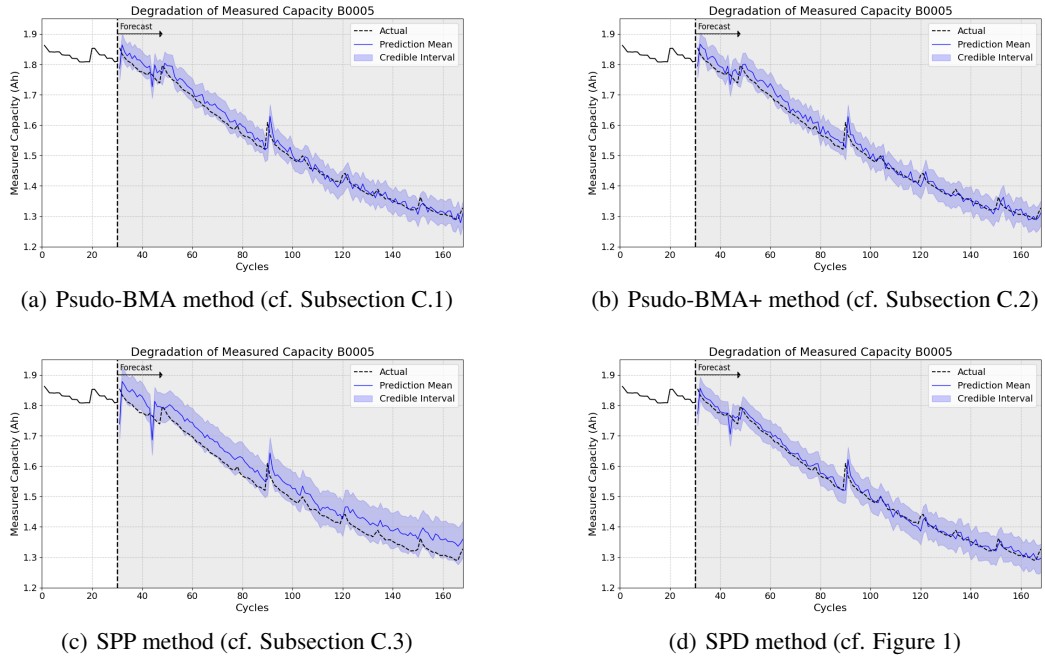

(a) Psudo-BMA method (cf. Subsection C.1)  (b) Psudo-BMA+ method (cf. Subsection C.2)

(c) SPP method (cf. Subsection C.3)  (d) SPD method (cf. Figure 1)

Figure 2: Battery capacity degradation forecasting results.

# 4 Discussion

The proposed research work demonstrates that the stacking of predictive distributions based on a Bayesian framework improves the accuracy and robustness of predictions compared with pseudo-BMA, pseudo-BMA+ and point prediction stacking approach. However, before drawing definitive conclusions about the application of the proposed solution in real-world applications,

*Robustness*

Credible intervals indicate the uncertainty associated with the data and the model (see Figure 2). Enhancing robustness involves reducing these intervals by minimizing uncertainty. This can be achieved by increasing observations to refine model uncertainty and using priors, such as maximum entropy or weakly informative priors, to tighten credible intervals.

*Scalability*

To analyze larger fleets of batteries, clustering similar batteries or using hierarchical modeling with a global model and specific group models would be more efficient than leave-one-out methods, enabling better data diversity and scalable adaptations.

# 5 Conclusion

This work presented a novel probabilistic fusion approach for battery state-of-health prognostics, combining BCNNs with Bayesian ensemble stacking techniques. The proposed stacking of predictive distributions method demonstrated superior performance on the NASA battery dataset, particularly in terms of probabilistic metrics. Results showed improved accuracy and well-calibrated uncertainty estimates compared to alternative probabilistic fusion methods, highlighting the potential of this approach for robust battery health monitoring.

The main contribution lies in the effective integration of model diversity and uncertainty quantification, enabling more reliable decision-making in battery management systems. However, the generalization to diverse battery types and operational conditions requires further investigation. Despite this limitation, the proposed method represents a significant progress in probabilistic battery prognostics, with potential applications in other health monitoring domains.

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

## A    Bayesian Convolutional Neural Networks (BCNN)

BCNN models are a Bayesian extension of the classical Convolutional Neural Network (CNN) models to include uncertainty associated with parameter estimation. This requires modification of the classical backpropagation algorithm through Bayesian techniques. The incorporation of uncertainty into the model is achieved by treating weights as random variables and applying Variational Inference to approximate posterior distributions. This results in a more robust model that predicts the complete probability density function (PDF).

BCNN models have been selected to improve the robustness and accuracy of model prediction with respect to classical CNN models. To this end, BCNNs make use of probabilistic distributions to model parameters and the uncertainty related to their training process, and prior distributions to incorporate previous knowledge, generate uncertainty estimations and mitigate over-fitting Blundell et al. (2015). In contrast, the classical learning models, e.g. non-Bayesian CNN models, focus on maximum likelihood estimation (MLE) and they overlook prior and posterior distributions. This leads to increasing error and decreasing model robustness in high uncertainty contexts, e.g. out-of-distribution data or manufacturing drifts.

The architecture of the BCNN models is shown in Figure 3 and it is defined as follows:

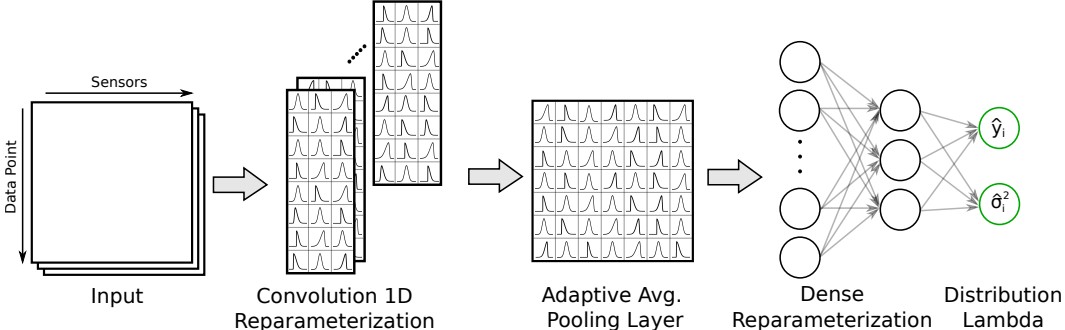

Figure 3: Schematic of the Bayesian convolution neural network.

- Input data: the input data for the BCNN is structured in a tensor format. The rows represent data samples of discharge cycles, and columns that correspond to features, such as the voltage and temperature over time. Notably, the input does not include the current discharge as it remains constant in this scenario.

- Convolutional 1D Reparametrization: this layer creates a convolution kernel that is applied to the input data. During the forward pass, kernel and bias parameters are drawn from a Gaussian distribution. It uses the reparameterization estimator to approximate distributions through Monte Carlo trials, integrating over the kernel and bias.

- Global Average Pooling 1D: this layer performs average pooling specifically for temporal data. It reduces the spatial dimensions of the input data to a single value per channel by calculating the average over the temporal dimension.

- Flatten: this layer reshapes input data into a one dimensional array, enabling compatibility between Bayesian convolutional layers and Bayesian dense layers.

- Dense Reparameterization: this layer implements a reparameterization estimator for Bayesian variational inference. It implements a stochastic forward pass via sampling from the kernel and bias distributions. This approach improves the robustness of the model, allowing uncertainty estimation in parameter values and supporting probabilistic modeling in deep learning.

- Distribution Lambda: this layer is responsible for producing the final results given the inputs and the learned weights from the previous layers. The output layer consists of two neurons representing the mean, $\hat{y}$ and variance, $\hat{\sigma}^2$, in order to quantify the expected value and its associated uncertainty. To ensure a positive variance, the neuron is activated using an exponential function.

## B  Dataset description

The effectiveness of the proposed method has been tested using a battery dataset from the NASA Ames Prognostics Center of Excellence (Saha and Goebel, 2007).

A subset of available battery data has been selected, focusing on batteries #5, #6, #7 and #18. Each battery is operated under various conditions including charging, discharging, and impedance analysis. Throughout the charge and discharge cycles, temperature, current, and voltage were meticulously recorded. During charging, a constant current mode at 1.5 A was maintained until the voltage reached 4.2 V, followed by a switch to constant voltage mode until the current dropped to 20 mA. Discharge cycles involved a constant load mode at 2 A until the voltage levels reached 2.7 V, 2.5 V, 2.2 V and 2.5 V for batteries #5, #6, #7 and #18, respectively. The experiment ended once the battery capacity decreased by 30%. These batteries had a maximum capacity of 2Ah with an end-of-life capacity set at 1.4Ah.

Figures 4(a), 4(b) and 4(c) show the evolution of voltage, current (constant), and temperature measurements with the increment of discharge cycles for the battery #5.

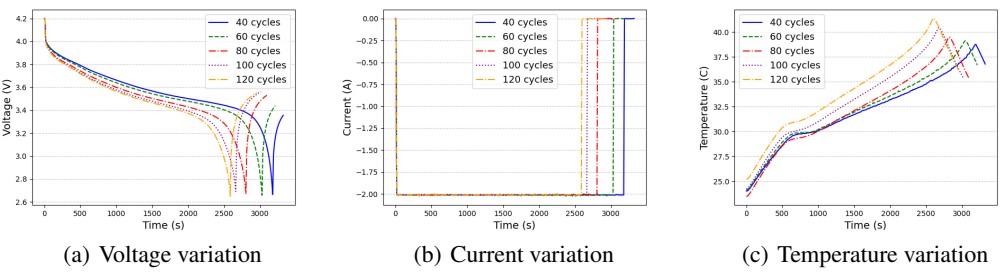

(a) Voltage variation  (b) Current variation  (c) Temperature variation

Figure 4: Feature variations due to an increasing number of discharge cycles in battery #5.

Figure 5 shows variations in capacity degradation rates for identical batteries. This is an indicator of uncertainty inherent in the manufacturing process, which affects SOH estimates.

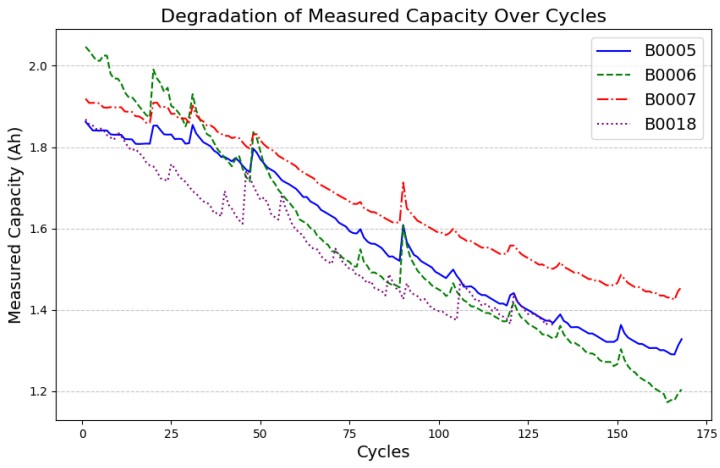

Figure 5: Capacity degradation data of Li-ion batteries.

## C Benchmarking

To compare the proposed stacking approach with alternative strategies, additional ensemble methods were implemented. These include pseudo-BMA, Pseudo-BMA+, and stacking of point predictions using the energy score as a scoring rule.

### C.1 Pseudo Bayesian Model Averaging

Pseudo Bayesian Model Averaging (Pseudo-BMA) is an approach that broadens the range of models under consideration by deriving model weights from estimated out-of-sample predictive performance. Pseudo-BMA supports the identification of the generalization error in each model and accounts for uncertainty across possible models by assessing performance on new or future data.

Yao et al. (2018) introduced Pseudo-BMA, which estimates model weights by renormalizing the expected log pointwise predictive densities (ELPD) of each candidate model. ELPD estimates the expected predictive performance of a model $k$ on new data, $\tilde{y}$, and is typically approximated using cross-validation techniques, such as leave-one-out (LOO) cross-validation, $ELPD_{loo}$. Yao et al. (2018) recommend using Pareto-smoothed importance sampling (PSIS) estimates of LOO cross-validation (PSIS-LOO) (Vehtari et al., 2017) to approximate $ELPD_{loo}$. This can be represented mathematically as:

$$
\begin{aligned}
\text{ELPD}_{\text{loo}}^k &= \sum_{i=1}^{N} \log p(y_i \mid y_{-i}, M_k) \\
&= \sum_{i=1}^{N} \log \int p(y_i \mid \theta_k, M_k) p(\theta_k \mid M_k, y_{-i}) \, d\theta_k \\
&\approx \sum_{i=1}^{N} \log \left( \frac{\sum_{s=1}^{S} r_{ik}^s p(y_i \mid \theta_k^s, M_k)}{\sum_{s=1}^{S} r_{ik}^s} \right) \\
&= \widehat{\text{ELPD}}_{\text{psis-loo}}^k
\end{aligned}
\tag{4}
$$

In this equation, $p(y_i|y_{-i})$ represents the LOO predictive density of the $i$-th data point, given data excluding that point. $p(y_i|\theta_k^s, M_k)$ is the predictive density of the $i$-th data point from the $s$-th sample of the posterior distribution of parameters $\theta_k$ and model $M_k$. To approximate the LOO predictive density, a corrective importance sampling weight $(r_i k)$ is applied to the vector of log predictive densities, estimated robustly using PSIS (Vehtari et al., 2017).

The resulting vector of ELPD values across models can be transformed into a set of weights using the softmax function:

$$
w_k = \frac{\exp(\widehat{\text{ELPD}}_{\text{psis-loo}}^k)}{\sum_{k=1}^{K} \exp(\widehat{\text{ELPD}}_{\text{psis-loo}}^k)}
\tag{5}
$$

### C.2 Pseudo Bayesian Model Averaging with Boostraping (Pseudo-BMA+)

An advanced approach to computing uncertainties related to LOO estimation involves using the Bayesian bootstrap (BB) method, as described by Vehtari and Lampinen (2002). The Bayesian bootstrap, introduced by Rubin (1981), offers a straightforward non-parametric approximation to any probability distribution. In this method, given samples $(z_1, \ldots, z_n)$ from a random variable $Z$, it is assumed that the posterior probabilities for all observed $z_i$ follow a Dirichlet$(1, \ldots, 1)$ distribution. Unobserved values of $Z$ are assigned zero posterior probabilities. Each BB replication generates a set of posterior probabilities $\alpha_{1:n}$ for all observed $z_{1:n}$, which can be represented as:

$$
\alpha_{1:n} \sim \text{Dirichlet}(\underbrace{1, \ldots, 1}_{n}), \quad P(Z = z_i|\alpha) = \alpha_i.
\tag{6}
$$

This leads to one BB replication of any statistic $\phi(Z)$ of interest:

$$\hat{\phi}(Z|\alpha) = \sum_{i=1}^{n} \alpha_i \phi(z_i). \tag{7}$$

The distribution over all replicated $\hat{\phi}(Z|\alpha)$ (generated by repeated sampling of $\alpha$) produces an estimation for $\phi(Z)$.

Given that the distribution of $\widehat{\text{ELPD}}_{\text{loo},i}^{k}$ is often highly skewed, BB is likely to perform better than a Gaussian approximation. In our model weighting approach, we define:

$$z_i^k = \widehat{\text{ELPD}}_{\text{loo},i}^{k}, \quad i = \{1, \ldots, n\} \tag{8}$$

We then sample vectors $(\alpha_{1,b}, \ldots, \alpha_{n,b})_{b=1,\ldots,B}$ from the $\text{Dirichlet}(\underbrace{1, \ldots, 1}_{n})$ distribution and compute the weighted means:

$$\bar{z}_b^k = \sum_{i=1}^{n} \alpha_{i,b} z_i^k. \tag{9}$$

A Bayesian bootstrap sample of $w_k$ with size $B$ is then calculated as:

$$w_{k,b} = \frac{\exp(n\bar{z}_b^k)}{\sum_{k=1}^{K} \exp(n\bar{z}_b^k)}, \quad b = \{1, \ldots, B\}, \tag{10}$$

Finally, the adjusted weight of the model $k$, which we term the Pseudo-BMA+ weight, is computed as follows:

$$w_k = \frac{1}{B} \sum_{b=1}^{B} w_{k,b}, \tag{11}$$

## C.3 Stacking of Point Prediction

An effective method for determining the weight of each model in the stacking process is by minimizing the leave-one-out mean squared error with a $L_2$ regularization term, $\lambda_{reg}$. The purpose of this term is to penalize large weights to preventing overfitting and balance individual model contributions. The weights are obtained through the following optimization problem:

$$\hat{w} = \arg\min_{w} \sum_{i=1}^{n} \left( y_i - \sum_{k=1}^{K} w_k \hat{f}_K^{(-i)}(x_i) \right)^2 + \lambda_{reg} \sum_{k=1}^{K} w_k^2 \tag{12}$$

where $\hat{f}_K^{(-i)}(x_i)$ represents the predicted value of the $k$-th model, when the $i$-th observation is left out of the training set. The regularization parameter, $\lambda_{reg}$, controls the strength of the applied regularization. To ensure a feasible solution, the weights are restricted to $w_k \geq 0$ and $\sum_{k=1}^{K} w_k = 1$.

Accordingly, the stacking of point prediction approach is defined as follows:

$$\hat{y} = \sum_{k=1}^{K} \hat{w}_k f_k(x|\theta_k) \tag{13}$$

where $\hat{y}$ represents the prediction of the ensemble for the test battery capacity, $\hat{w}_k$ denotes the weight assigned to the $k$-th battery base model, and $f_k(x|\theta_k)$ is the prediction made by the corresponding base model ($\text{BCNN}_k$).

# D   Performance Assessment Metrics

The accuracy of the regression is measured by mean squared error (MSE), while negative log likelihood (NLL) assesses model performance by quantifying prediction probabilities. Finally, The correctness of probability predictions is assessed through the continuous ranked probability score (CRPS).

## D.1   Continuous Ranked Probability Score (CRPS)

CRPS can be formally expressed as a quadratic measure of discrepancy between the predicted Cumulative Distribution Function (CDF), $F(\cdot)$, and the observed empirical CDF for a given scalar observation $y$ (Zamo and Naveau, 2018):

$$CRPS(F, y) = \int (F(x) - \mathbb{I}(x \geq y_i))^2 dx, \tag{14}$$

where $\mathbb{I}(x \geq y_i)$ is the indicator function, which models the empirical CDF.

To obtain a single score value from Eq. (14), a weighted average is calculated for each individual observation of the test set (Gneiting et al., 2005):

$$CRPS = \frac{1}{N} \sum_{i=1}^{N} CRPS(F_i, y_i) \tag{15}$$

where $N$ denotes the total number of predictions.

## D.2   Negative Log Likelihood (NLL)

NLL metric assesses probabilistic models by using the likelihood concept, which indicates how likely the observed data is given model parameters (Bosman and Thierens, 2000). Likelihood ($\mathcal{L}$) is the product of each observation's probability density function (PDF), expressed mathematically as

$$\mathcal{L}(\theta \mid X) = \prod_{i=1}^{N} f(x_i|\theta) \tag{16}$$

where $\theta$ denotes model parameters and $X$ includes $N$ data points. NLL is preferred for optimization since minimizing NLL is equivalent to maximizing the log-likelihood, facilitating the discovery of model parameters that best explain the observed data, represented by

$$-\log \mathcal{L}(\theta \mid X) = -\sum_{i=1}^{n} \log f(x_i \mid \theta) \tag{17}$$

