# OpenReview forum: "Probabilistic Fusion Approach for Robust Battery Prognostics"
_NeurIPS.cc/2024/Workshop/BDU — NeurIPS BDU Workshop 2024 Poster_

### Official Review · Reviewer_2Kyb · 2024-10-04
**Solid Submission with some work needed in data application**

**Rating:** 7
**Confidence:** 3

**Review:**

This paper demonstrates a novel combination of bayesian model averaging and bayesian convolutional neural networks in order to better model battery health over time. Care has been made to take advantage of the uncertainty estimate that bayesian neural networks give and a robust model averaging approach using multiple types of model averaging has been performed.

While the application section, which demonstrates the performance of the algorithm on a publicly available dataset seems well done, there are two points that could do with improvement. First, the citation for the dataset "Saha, B. and Goebel, K. (2007). Nasa ames prognostics data repository. NASA Ames, Moffett Field, CA, USA." points to the location of all prognosis datasets, but the site itself has multiple battery datasets and each gives their own separate citation. For replicability, please update the citation to the more specific dataset. Second, the model comparisons and evaluations were not compared to any external benchmarks so it is difficult to know how the performance of this approach compares to existing methods.

However, overall, I believe the paper is well written, interesting, and well demonstrates bayesian decision making using AI.

---

### Official Review · Reviewer_Bn12 · 2024-10-08
**Accept**

**Rating:** 6
**Confidence:** 4

**Review:**

In this work, the authors develop a probabilistic model fusion approach for battery state-of-health predictions. The proposed approach combines Bayesian convolutional neural networks and ensemble methods to reliably predict uncertainty. The novelty of the approach is relatively limited. It is unclear what the core differences are from Alcibar et al. (2024). However, I think the core contribution is the successful application to battery SOH, which is interesting and non-trivial.

Minor suggestions:
Please add abstract before introduction.
Please emphasize and discuss the difference and similarity between the proposed method and existing ones.

---

### Decision · Program_Chairs · 2024-10-09

Accept (Poster)